# Analysis of Mycotoxins Contamination in Poultry Feeds Manufactured in Selected Provinces of South Africa Using UHPLC-MS/MS

**DOI:** 10.3390/toxins11080452

**Published:** 2019-08-02

**Authors:** Sharon Maphala Mokubedi, Judith Zanele Phoku, Rumbidzai Naledi Changwa, Sefater Gbashi, Patrick Berka Njobeh

**Affiliations:** 1Department of Biotechnology and Food Technology, Faculty of Science, University of Johannesburg, P.O. Box 17011, Doornfontein Campus, Johannesburg, South Africa; 2Agricultural Research Council, Onderstepoort Veterinary Institute, Toxicology and Ethnoveterinary Medicine, Public Health and Zoonoses, Private Bag X05, Onderstepoort, Pretoria North 0110, South Africa

**Keywords:** mycotoxins, poultry feeds, UHPLC-MS/MS, South Africa

## Abstract

A total of 105 different types of poultry feed samples from South Africa were simultaneously analysed for the presence of 16 mycotoxins using ultra-high-performance liquid chromatography coupled to a triple quadrupole mass spectrometer (UHPLC-MS/MS). The data revealed the presence of 16 mycotoxins in the various poultry feed samples. Fumonisin B_1_ (FB_1_) was the most dominant recovered from 100% of samples analysed at concentrations ranging between 38.7 and 7125.3 µg/kg. This was followed by zearalenone (ZEN) (range: 0.1–429 µg/kg) and deoxynivalenol (DON) (range: 2.5–154 µg/kg). Samples were also found to be contaminated with fumonisin B_2_ (FB_2_) (range: 0.7–125.1 µg/kg), fumonisin B_3_ (FB_3_) (range: 0.1–125.1 µg/kg), α-zearalenol (α-ZEL) (range: 0.6–20 µg/kg ), β-zearalenol (β-ZEL) (range: 0.2–22.1 µg/kg), 3-acetyldeoxynivalenol (3-ADON) (range: 0.1–12.9 µg/kg) and 15-acetyldeoxynivalenol (15-ADON) (range: 1.7–41.9 µg/kg). *Alternaria* mycotoxin, i.e., Alternariol monomethyl ether (AME) was recovered in 100% of samples at concentrations that ranged from 0.3–155.5 µg/kg. Aflatoxins (AFs) had an incidence rate of 92% with generally low concentration levels ranging from 0.1–3.7 µg/kg. Apart from these metabolites, 2 type A trichothecenes (THs), i.e., HT-2 toxin (HT-2) (range: 0.2–5.9 µg/kg) and T-2 toxin (T-2) (range: 0.1–15.3 µg/kg) were also detected. Mycotoxin contamination in South African poultry feed constitutes a concern as correspondingly high contamination levels, such as those observed herein are likely to affect birds, which can be accompanied by severe health implications, thus compromising animal productivity in the country. Such exposures, primarily to more than one mycotoxin concurrently, may elicit noticeable synergistic and or additive effects on poultry birds.

## 1. Introduction

In basic terms, animals require an adequate supply of carbohydrates, proteins, fats/oils, vitamins, minerals, and water. However, the composition of raw materials in animal feeds varies from formulation to formulation and between different species [1]. In South Africa, the poultry industry is divided into two, i.e., the broiler industry, which supplies poultry meat, and the egg industry that is generally layer and breeder farm chickens for egg production [2]. While poultry feeds are largely composed of cereal grains, protein, and vitamin supplements [3], the most significant ingredients used in South African poultry feeds are maize, oilcake, soybeans and fishmeal [1]. The poultry industry in the country remains the second largest consumer of maize often used as the main ingredient in poultry feed formulation [2].

In most cases, feeds are produced and stored for a long period of time before distribution and this may endanger the quality for animal consumption. Poor composition of raw materials, high carbon, and moisture in feeds may very often lead to fungal and mycotoxin contamination [4,5,6,7]. Mycotoxins are well-known toxic secondary metabolites mainly produced by various filamentous fungal species of the *Aspergillus*, *Fusarium*, *Penicillium*, and *Alternaria* genera, which are commonly recognised as contaminants of food and feedstuffs [7,8]. Even though over 300 mycotoxins are known to occur under a wide range of climatic conditions, those that are significant due to health and economic significance in sub-Saharan Africa (SSA) are the aflatoxins (AFs), fumonisins (FUMs), in addition to ochratoxins (OTs) and trichothecenes (TH) [8,9]. The significance of these mycotoxins rests on their existence in food and feed above regulatory limits. Other than their presence in food and feed, emerging *Alternaria* mycotoxins such as alternariol monomethyl ether (AME), alternariol (AOH) and tenuazonic acid (TeA) are frequently encountered as contaminants of feed ingredients [10] that could occasionally find themselves in the final product. Although their contamination levels are generally low with limited toxicity data in chickens, their health risk cannot be completely excluded when noting their possible reproductive and immune system effects in both humans and animals [11,12,13].

Most cases related to chronic conditions result from the consumption of mycotoxins regularly at low doses, which go unnoticed especially since they can co-occur throughout the animal’s developmental cycle, causing multiple diseases [3,14,15,16,17]. In poultry, the consumption of ochratoxin A (OTA) contaminated feed can result in ochratoxicosis, a general term for OTs causing disease, characterised by low egg production and poor weight gain [3,5], whereas the consumption of AF contaminated feed or feedstuff results in the hepatic condition, aflatoxicosis, causing anorexia and affecting growth rates, decreasing egg production and increasing death rates [3,17]. Aflatoxins, which mostly flourish under tropical conditions, have the ability to cause extensive liver damage in poultry and other livestock [5]. The co-occurrence of FUMs, deoxynivalenol (DON), and zearalenone (ZEN) in poultry feed can possibly lead to symptoms such as reduced villus height in broiler chicks [14,16].

Feed represents one of the main aspects in poultry production. Both broilers and layers are effective in converting feed to food products [15,18] and as such, there are high chances of the carry-over of mycotoxins into edible by-products from poultry bird fed with contaminated feeds. The situation of poultry feeds contaminated by mycotoxins is an eminent potential of carry-over affecting negatively poultry health and performance. Broiler and layer chickens in farms affected by mycotoxicosis have been seen to exhibit symptoms such as weight loss, reduced feed conversion competency, immunosuppression, failure to vaccination responses, low fertility, high chances of egg blood spots, kidney enlargement, pale fatty liver, gizzard erosions, increased incidence of leg malformations, visceral haemorrhages, inclusion body hepatitis and oral lesion [7,15,18]. Such conditions may negatively and seriously affect the poultry industry.

In South Africa, few studies on mycotoxins in poultry feed are reported, meaning that limited studies in monitoring them exist. Recently, a study on the dairy cattle feeds from the Gauteng Province of South Africa established the occurrence of 15 mycotoxins [19]. A study on compound feeds produced in South Africa reported FUMs as the most dominant mycotoxin that co-existed with ZEN and DON in 67% of all the samples analysed [8]. The survey on South African feed samples revealed the occurrence of 75% FUMs, 90% DON, 20% ZEN and 14% AFs [20]. The nature and presence of mycotoxins in South African animal feeds should be an on-going matter since some of these mycotoxins are found in animal products from animals that consume feeds contaminated with mycotoxins [21,22]. Despite an increased interest in mycotoxins contamination and their health effects, only a few studies covered the scope of mycobiota and mycotoxins in poultry feed in the country [8,23,24].

Hence in this study, the most commonly encountered mycotoxins (AFs, FUMs, DON, ZEN, T-2 toxin and HT-2 toxin) including the *Alternaria* toxin, AME and derivatives of ZEN i.e., α-ZEL and β-ZEL, and derivatives of DON i.e., 3-ADON and 15-ADON in poultry feed manufactured in South Africa were screened for multiple contamination using UHPLC-MS/MS. This was particularly important since the co-occurrence of these mycotoxins could synergistically act with one another to potentiate additional or at best synergistic toxic effects in animals, as reported elsewhere [5,14].

## 2. Results

### 2.1. Mycotoxin Identification

The present study analysed 16 different mycotoxins in various poultry feed from some selected manufacturing sites as described subsequently in Section 5. Associated method validation parameters are presented in Table 1. The linearity coefficient of determination ranged from 0.9919 to 0.9999, while the apparent recoveries for 16 mycotoxins extracted from each spiked matrix ranged from 73 to 132.3%. Accordingly, 16 mycotoxins (raw data presented in Appendix A
Table A3) were simultaneously analysed based on a multi-mycotoxin dilute and shoot method using UHPLC-MS/MS. The employed analytical method allowed for the accurate analysis of the targeted mycotoxins based on retention profiles and selected MS parameters in Table A2 (Appendix A).

### 2.2. The Incidence Levels of Mycotoxins in Poultry Feed

The findings in the present study show that commercial poultry feeds were contaminated with multiple mycotoxins. In total, 16 mycotoxins were recovered, with 50% of the analysed samples contaminated with all 16 mycotoxins as presented in Appendix A
Table A3 (raw data). Although sample size amongst the different feeds analysed was not even, the incidence of 16 mycotoxins were observed in 63% of the broiler layer feed, and in 57% of both the starter and the farmix feed. The frequencies of all 16 mycotoxins in feed for breeders, growers, and finishers were in 47, 39, and 27%, respectively. A combination of significant mycotoxins, i.e., AFs, FUMs, ZEN, T-2, HT-2, DON, AME and alongside some of their metabolites (α-ZEL, β-ZEL, 3- and 15-ADON) co-occurred in individual poultry feed samples in this study. A summary of selected mycotoxins occurring naturally singly or in combination in the analysed samples is represented in Table 2. The high occurrence of FB_1_ followed by ZEN and DON dominated in all provinces in comparison to other mycotoxins is summarised in Table 3.

The overall data revealed that all the 105 poultry feed samples analysed contained multiple mycotoxins. Figure 1 re-presents the co-occurrence of the groups of mycotoxins commonly detected in poultry feeds, which are AFs, FUMs, ZENs, DONs, HT-2 and T-2. This was achieved by calculating the frequencies of all the possible co-occurrence patterns of the mycotoxins in the samples. The highest co-occurrence of mycotoxins that appeared in the poultry feed samples was AFs, FUMs, ZENs and DONs which occurred with a frequency of 51%, while the co-contamination of FUMs, ZENs and DONs occurred in 42% of the samples. A similar frequency of 26% was detected with samples concurrently contained AFs + FUMs and ZENs, and AFs + FUMs + HT-2 and T-2. In general, out of 17 mycotoxins analysed in this study, 10 mycotoxins (ZEN, α-ZEL, β-ZEL, FB_1_, FB_2_, FB_3_, AFB_2_, AME, T-2 and HT-2) were individually detected in all of the poultry feed samples analysed; this excludes DON, 3-ADON, AFG_1_, AFB_1_, AFG_2_ and 15-ADON (Appendix A raw data Table A3). 

## 3. Discussion

Poultry feed was analysed for mycotoxin contamination. The overall results of the reported mycotoxins in poultry feeds analysed herein are fairly in line with those of compound feeds previously analysed by Njobeh and colleagues [8], suggesting that AFs, FUMs, DON, and ZEN are the most common contaminants of poultry and other livestock feeds produced in South Africa; furthermore, reporting FB_1_ with higher concentration than other mycotoxins. Although the *Fusarium* toxins as such are predominately found as common contaminants of foods and feedstuffs mostly in maize-based products [8,9], the study on compound feeds established the high levels of FUMs in the cattle feeds and suggested that the concentration was not linked to the levels of maize as poultry feed remains the biggest consumer of maize [8]. In this study, FB_1_, FB_2_, and FB_3_ were recovered in all poultry feed samples. Fumonisin B_1_ was recovered from feeds at a maximum value of 7125.3 µg/kg as compared to FB_2_ and FB_3_ with maximum values of 125.1 and 115.1 µg/kg, respectively. These values are comparably higher than those previously reported in other animal feed studies conducted in the country [8,19]. The much higher levels of FUMs established in this study are most likely to be due to the increased inclusion of increased levels of fibre and of protein content which may be indicators of certain ingredients in different poultry feed groups (Appendix A
Table A1).

Higher concentrations of FB_1_ compared to FB_2_ and FB_3_ are very often reported [8,10]. The natural occurrence of FUMs in maize and maize-based products in several African countries including South Africa is reported with these products as principal substrates [9,25]. Whereas this study established that the high contamination levels of FUMs are attributable to increased FB_1_ contamination (being at least 40 times higher than other analysed FUMs analogue), however, all the observed levels were below that regulated by the South African Department of Agriculture, Forestry and Fisheries (DAFF) set at 50,000 µg/kg in poultry feed [26]. Furthermore, the FUMs levels investigated herein were demonstrated to be well within the European Commission (EC) guidance values set at 20,000 µg/kg for combined maximum levels of FB_1_ and FB_2_ in complementary and complete feeds for poultry [27]. The FUMs levels were also found to be well within the US Food and Drug Administration (USFDA) guidance levels for the sum of FB_1_, FB_2_ and FB_3_ at 30,000 µg/kg for breeding poultry and 100,000 µg/kg in feed for poultry being raised for slaughter [27].

Zearalenone and its derivatives strongly interfere with animal reproductive systems and decreased fertility [16]. Whereas currently both DAFF and USFDA do not have regulatory limits for ZEN in poultry feeds [26,27], this mycotoxin is gradually being recognized as a significant contaminant of cereal crops [16,20] and it is regulated by the EC with guidance values of 2000 µg/kg for cereal and cereal-based products and 3000 µg/kg for maize products [28]. The occurrence of ZEN and its derivatives have been reported in maize and maize-products but at low concentrations [22]. Lower levels of mycotoxin derivatives are common due to possible biotransformation of the conjugated mycotoxin from the parent mycotoxin [5]. In South African studies, ZEN is reported in lower concentrations, which makes it less problematic and probably why it is not regulated in South African poultry feed [8,20]. In a recent South African study on multi-mycotoxins occurrence in Gauteng cattle feeds, ZEN was established in 60% samples at a mean concentration of 2.8 µg/kg [19], whereas Njobeh et al. [8] reported ZEN in chicken feeds with an incidence rate level of close to 52% at a maximum value of 610 µg/kg; the levels of this toxin in our study were found in 100% tested samples to be as high as 429 µg/kg. The derivatives of ZEN were also detected in 100% of the samples but much lower concentrations were recovered (maximum values of 22.1 µg/kg for β-ZEL and 20 µg/kg for α-ZEL). Although they are within the EC guidance values for both cereal and maize-based products, such high levels of ZEN and its derivatives are not common in SSA. However, the world mycotoxins survey showed ZEN as the third major contaminant following DON and FUM in animal feed samples from African countries including South Africa [25]. 

Trichothecenes (THs) are *Fusarium*-derived toxins with a high potency that are commonly associated with *Fusarium* head blight in cereal grains [16]. Those of significance in this study were the type A TH, i.e., T-2 and HT-2 and commonly occurring type B TH DON along with its acetylated derivatives, i.e., 15-ADON and 3-ADON. This group of mycotoxins is very common and problematic in European countries, thus considered as significant contaminants in colder European climates [9]. Hence, the EC recently placed recommended levels of 500 µg/kg for the sum of T-2 and HT-2 in cereal-based products and 250 µg/kg in compound feeds [29]. There is inadequate data on TH toxins in South Africa. None of the South African feed samples analysed previously contained either T-2, HT-2, or both, hence, there is no room for comparison on the occurrence of TH toxins in South Africa. Thus, to the best of our knowledge within the EC guidance values, this study reports for the first time, T-2 with a range of 0.1–15.3 µg/kg and for the second time following a recent study on dairy cattle feed contamination [19], our study reports HT-2 (range of 0.2–5.9 µg/kg) in 100% of the poultry feed samples analysed. 

The incidence of DON has consistently been reported in South African crop-based products and for that reason, the mycotoxin has been of great interest in the country [8]. Department of Agriculture, Forestry and Fisheries of South Africa regulates this mycotoxin in poultry feed limiting the level to 4000 µg/kg [26]. In South African feedstuffs, Njobeh et al. [8] found DON in chicken feeds with mean levels of 620 ± 386 µg/kg at a maximum level of 1980 µg/kg while Changwa et al. [19] reported lower levels of DON with a maximum value of 81.6 µg/kg in dairy feeds. Our study reports DON incidence of 99% but at comparatively much lower concentrations (max: 154 µg/kg) than those previously reported [8,19]. In this study, derivatives of DON, i.e., 3-ADON and 15-ADON (max: 12.9 and 44.9 µg/kg, respectively) were simultaneously analysed in combination, taking into consideration their isometric and co-eluting nature. The incidence levels of DON and its derivatives in this study remained well within the DAFF’s maximum regulatory limits of 4000 µg/kg for poultry feeds [26].

Once more, this study established the natural occurrence of AME (max: 155.5 µg/kg) in 100% of analysed poultry feed samples, the *Alternaria* toxin. There are no regulations currently available for this toxin in food or feed, however, studies have focused on establishing its profile along with alternariol (AOH), both being the main toxins produced by *Alternaria alternata* most commonly encountered in fruit products, sunflower seeds, wheat and other agricultural products [10,11,12]. The presence of *Alternaria* toxins has been reported in maize samples [10,11] and in sunflower seeds ranging between 1 and 103 µg/kg [12]. There is scarce data concerning levels of this toxin in toxicological studies, hence, no regulations are set currently.

Results on AFs revealed unusual occurrence patterns wherein higher concentrations of the less potent AFG_1_ and AFG_2_ (mean: 0.7 and 0.5 µg/kg, respectively) than those of AFB_1_ and AFB_2_ (mean: 0.2 and 0.4 µg/kg, respectively) and moreover, the G-types occurred more frequently than the B-types as observed. In animal feed, similar findings were reported [30,31,32], whereas, in raw maize and groundnut samples from Malawi lower mean concentrations of AFB_1_ than that of AFG_2_ were also reported [31]. The proportion of B-types and G-types AF are highly influenced by an ecological niche of the parent fungus [30,31]. *Aflatoxin flavus* is known to be an AFB_1_ and AFB_2_ producer, while *A. parasiticus* produces all four AFs (AFG_1_, AFG_2_, AFB_1_, and AFB_2_) [17,30,31]. Such unusual changes in AFs pattern may be related to climate change and global warming that could impact and compromise the behaviour of fungal plant pathogens [33]. None of the feed samples analysed was found at higher contamination level above the South African regulatory limit of 20 µg/kg, the USFDA limit range of 10–30 µg/kg, and the EU regulatory limit of 2 µg/kg for AFs in poultry feeds [26,27].

The data demonstrated the co-occurrence of analysed metabolites at various concentrations. Mycotoxins such as DON and ZEN are both produced by *Fusarium* species found under cool and wet conditions [9]. Mycotoxins such as FUMs, DON and ZEN, produced by *Fusarium* spp. are most frequently encountered in feeds and feedstuffs [4,8,34,35,36]. In this study, the overall data revealed mycotoxins co-contamination of poultry feed samples (Figure 1). Based on the results of this study, exposure of individual poultry feed samples to multiple mycotoxins simultaneously is highly likely as a result of AFs, FUMs, ZENs, and DONs, as we observed co-contamination of these mycotoxins with the a higher frequency of 51%. Mycotoxins can co-occur even at levels above regulatory limits very often in food and feed commodities causing negative health effects in human and animals.

A systematic review of over a hundred papers between 1987–2016, revealed 127 mycotoxin combinations, of which AFs + FBs, AFs + OTA, DON + ZEA, and FBs + ZEA were amongst the most frequently co-occurring combinations in cereal crop [27]. It has been reported that ZEA usually co-occurs with one or more of the THs, because of the ability of its producing fungi to synthesize more than one mycotoxin [16]. Mngadi et al. [23] provided some data in which several mycotoxins co-occur in feeds within the country. In our study, the co-existence of mycotoxins within the same sample was very common and data revealed that 91 samples contained multiple mycotoxins, with 67% that with 3 mycotoxins (FB + DON + ZEA), 26% with 4 mycotoxins (FB + DON + ZEA + AF).

Even though in this study, levels of the tested mycotoxins in individual samples were either below regulatory limits or at levels that could not elicit any toxic effect in poultry, a number of mycotoxins were found to co-occur with one another. Such co-occurrences of multiple mycotoxins may provoke some synergistic actions or additive effects thus, inducing various toxic effects in poultry [8]. The data reported in this study established that 51% of the samples were contaminated with multiple mycotoxins.

## 4. Conclusions

From our study, the levels of proteins in poultry feeds along with fibre content are indicators of certain ingredients which are most likely to be the reason behind the high values of mycotoxin contamination in poultry feed. In addition, some feeds are composed of raw materials, which are already contaminated, thus inevitably contributing to a high level of toxin contamination in formulated feeds. From the results obtained in this study, it can be concluded that poultry feeds are more contaminated with *Fusarium* mycotoxins (FUMs, ZENs, and DONs) as they were found at the highest contamination levels than other mycotoxins. The co-occurrence of multiple mycotoxins in poultry feed was noted and could pose some addictive or synergistic effects on the health and productivity of poultry. The unusual patterns of contamination of feeds by such mycotoxins as AFs, as well as the sudden appearance of AME, T-2, and HT-2 as established in this study justifies the need for feed manufacturers to monitor feed consignments for mycotoxins on a regular basis. Further to that, such findings may bring a shift in the implementation of management strategies for fungal and mycotoxin contamination of animal feeds in South Africa.

## 5. Materials and Methods

### 5.1. Sampling

The present study analysed different poultry feed manufactured in selected sites of South Africa, which included broiler starters, broiler growers, broiler finishers, broiler layer and broiler breeder and farmix feeds. This was mainly to assess the degree to which these agricultural products are naturally contaminated with mycotoxins. Poultry feed samples (105 in total) with an equivalent weight of ±500 g each from five provinces of South Africa in Figure 2, namely, Gauteng (45 samples), KwaZulu-Natal (26 samples), Eastern Cape (5 samples), Western Cape (17 samples), and North West (12 samples) were donated (between May and June 2015) by a member of the South African Animal Feed Manufacturers’ Association (AFMA). According to the donor, samples were randomly collected from each plant using a sampling spear from several spots within the lot and placed in sterile sealed plastic bags. The samples were further classified into six groups—broiler starter (7), broiler grower (18), broiler finisher (11), broiler breeder (15), broiler layer (8), and farmix (46) feeds. The raw material composition of the different poultry feed group is represented in Appendix A
Table A1. Samples were kept in cooler boxes, transported to the University of Johannesburg and stored immediately at −4 °C until analysed.

### 5.2. Reagents

Liquid Chromatography grade acetonitrile (ACN), methanol (MeOH) and analytical grade formic acid (purity > 98%) for organic solvents and mobile phases preparation were purchased from Sigma-Aldrich (Steinheim, Germany). Ultrapure deionized water was obtained from Millipore Milli-Q System (Merck, South Africa). Disposable Norm-Ject 10 mL syringe polypropylene and syringe filters 25 mm, 0.22 µm filter units (Restek, Japan) were also used.

#### 5.2.1. Mycotoxin Standards

The analytical standards of mycotoxins ZEN, α-ZEL, β-ZEL, FB_1_, FB_2_, FB_3_, AME, HT-2 and T-2 toxin were obtained from Sigma-Aldrich (Steinheim, Germany), AFB_1_, AFB_2_, AFG_1_, AFG_2_, DON, 3-ADON and 15-ADON were purchased from Trilogy^®^ (Washington, DC, USA).

#### 5.2.2. Stock and Working Standard Preparation

For experimental purposes, a combination of standard stock solution was freshly prepared for calibration purposes and spiking experiments on the least contaminated samples. Standard stock solutions were prepared into two groups: The AFs mix, which consisted of AFB_1_, AFB_2_, AFG_1_ and AFG_2_ in acetonitrile at a concentration of 250 µg/kg each, and the multi-mix group, also in acetonitrile consisted of FB_1_, FB_2_, FB_3_, ZEN, α-ZEL, and β-ZEL at individual concentrations of 625 µg/kg each; DON, 3-ADON, 15-ADON, AME, HT-2 and T-2 at 1250 µg/kg each. The working standard solutions were prepared by diluting the stock solutions with acetonitrile making five standard concentrations to calibrate the instrument and establish external calibration curves. 

### 5.3. Sample Preparation

One-hundred and five poultry feed samples were screened for multiple mycotoxins by UHPLC-MS/MS following the dilute and shoot method [37]. Ten grams of milled dried poultry feed per sample was homogenised in a freshly prepared 40 mL extraction solvent consisting of acetonitrile:water:formic acid (79:20:1, *v/v/v*) on a bench shaker (LABCON GmbH, Heppenheim, Germany) for 60 min at 180 rpm using horizontal bench shaker. The supernatant was then centrifuged (Eppendorf Millipore laboratory 5702R Centrifuge, Merck, South Africa) at 1358 *g* for 10 min. The extracts were collected and filtered by passing through 0.22 μm particle size PTFE syringe filter units, then injected into the UHPLC system. 

### 5.4. UHPLC-MS/MS Parameters and Analysis

Chromatography analysis was performed using a Shimadzu UHPLC 8030 equipment coupled to an MS (Shimadzu Corporation, Tokyo, Japan) instrument capable of obtaining 500 MRMs per sec with an ultrafast scan speed of 15,000 u/s, and a polarity switching of 15 s. The chromatographic separation was achieved on an LC-30AD Nexera, which was connected to a SIL-30 AC Nexera autosampler and a CTO-20 AC Prominence Column Oven. The oven was equipped with a Raptor^TM^ ARC-18 column from Restek (2.7 µm, 2.1 × 100 mm) (Restek Corporation, Garden City, Bellefonte, PA, USA). The column was maintained at a constant temperature of 40 °C. The elution solvents mobile phase consisted of A (0.1% formic acid in deionized water) and B (0.1% formic acid in acetonitrile: methanol (50:50, *v/v*)) was delivered at a constant flow rate of 200 µL/min. The elution gradient program had a total run time of 17 min and started with 10% B for 0.1 min, increased steadily to 95% B at 8.4 min, at which point it was kept constant for 3 min, and then the initial condition (10% B) was re-established for 1 min and the column allowed to re-equilibrate for 4.5 min for the next run. 

Following the chromatographic separation, analytes were committed to a Shimadzu triple quad mass spectrometry detector model 8030 (Shimadzu Corporation, Kyoto, Japan) for detection and quantification. The ionization source was an electro-spray ionization (ESI) operated in a positive mode at an event time of 0.206 s. Data was acquired by a multiple reaction monitoring (MRM) method at optimized MS conditions for the analytes (Table 1). The ionization source parameters were optimized, interface nebulizing gas flow rate was 3 L/min, desolvation (DL) temperature was 250 °C, heat block temperature was 400 °C, and drying gas flow rate was 15 L/min. 

### 5.5. Method Validation and Quantification in Poultry Feed

Validation of the method was carried out for all the mycotoxins tested [37]. For quantification purposes, external calibration curves were established based on serial dilutions of the multi-analyte standard solutions in the following ranges:(1)6.25, 12.5, 125, and 250 µg/kg for AFB_1_, AFB_2_, AFG_1_, and AFG_2_(2)31.25, 62.5, 625, and 1250 µg/kg for DON, 3-ADON, 15-ADON, AME, HT-2, and T-2(3)15.6, 31.25, 312.5, and 625 µg/kg for FB_1_, FB_2_, FB_3_, ZEN, α-ZEL, and β-ZEL

Linear calibration curves were considered satisfactory when correlation coefficients (R^2^) were greater than 0.99. The apparent recovery experiments were determined in triplicates on three least contaminated samples for external calibrations by spiking 5 g of each with 100 µL of multi-mycotoxin standards with a known concentration. Subsequently, spiked samples were mixed and kept in a fume cupboard at room temperature to establish the equilibrium between the sample matrix and the toxins. Spiked samples were extracted as described in Section 5.3 following overnight establishment. From each spiked sample, 5 µL of the extract was injected into the UHPLC system. Each analyte detected was quantified by comparing its peak area on the calibration plot of the equivalent mycotoxin standard. Limits of quantification (LOQ) and detection (LOD) (Results section Table 1) were estimated using the lowest concentrations in the spiked samples estimated at a signal-to-noise ratio (S/N) of 3:1 and 10:1, respectively. The apparent recovery percentage (%R) for 16 mycotoxins extracted from each spiked matrix were determined and calculated following the Equation (1): (1)$$\%R\;=\;\frac{\mathrm{mycotoxin}\;\mathrm{in}\;\mathrm{spiked}\;\mathrm{sample}\;-\;\mathrm{mycotoxin}\;\mathrm{in}\;\mathrm{not}\;\mathrm{spiked}\;\mathrm{sample}}{\mathrm{Spiked}\;\mathrm{mycotoxin}}\times 100$$

## Figures and Tables

**Figure 1 toxins-11-00452-f001:**
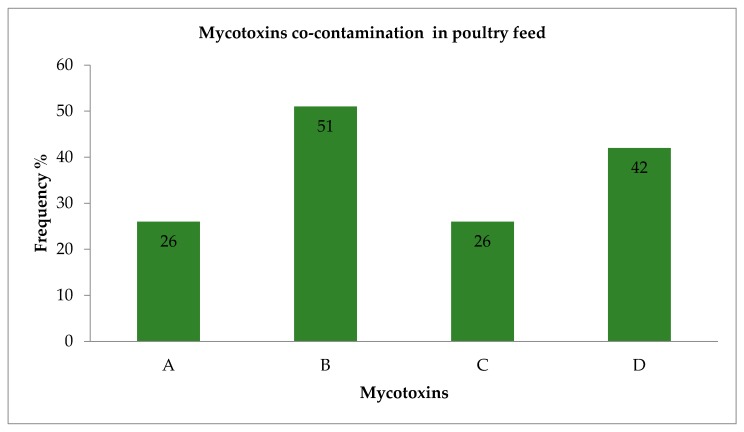
Mycotoxins co-contamination in poultry feed samples from selected provinces of South Africa. **A**: AFs + FUMs + ZENs (zearalenone and derivatives); **B**: AFs + FUMs + ZENs + DONs (deoxynivalenol and derivatives); **C**: AFs + FUMs + HT-2 + T-2; **D**: FUMs + ZENs + DONs at frequencies of 26, 51, 26 and 42%, respectively.

**Figure 2 toxins-11-00452-f002:**
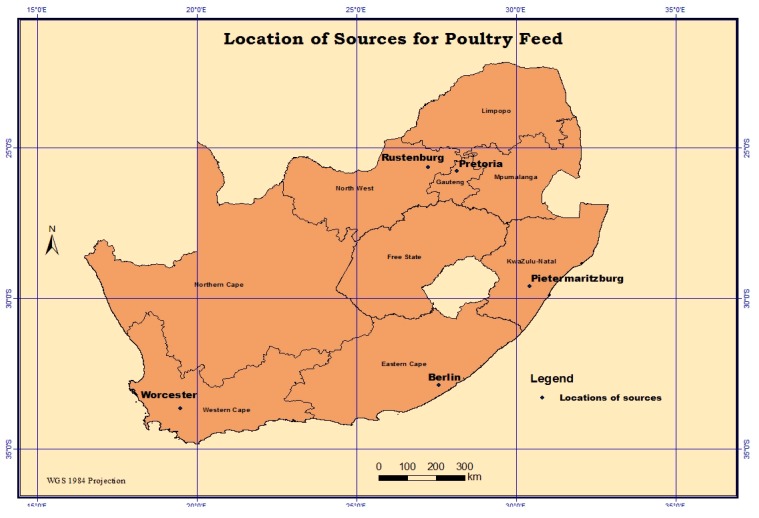
Poultry feed manufacturing sites in South Africa: Sampling sites are located in Pretoria (Gauteng Province), Rustenburg (North-West Province), Pietermaritzburg (KwaZulu-Natal Province), Berlin (Eastern Cape Province), and Worcester (Western Cape Province).

**Table 1 toxins-11-00452-t001:** Quantification parameters based on calibration curves. Data display affiliated method for limits of quantification (LOQ) and limits of detection (LOD) for each mycotoxin.

Compound	Calibration Level (µg/kg)	Spiking Concentrations (µg/kg)	Apparent Recovery (%)	LOQ (µg/kg)	LOD (µg/kg)
ZEN	15.6, 31.25, 312.5, 625	31.3	106.1	0.3	0.1
α-ZEL	15.6, 31.25, 312.5, 625	31.3	82.5	1.8	0.6
β-ZEL	15.6, 31.25, 312.5, 625	31.3	73	0.6	0.2
T-2	31.25, 62.5, 625, 1250	62.5	107.4	0.4	0.1
FB_1_	15.6, 31.25, 312.5, 625	31.3	115.8	63.9	19.4
FB_2_	15.6, 31.25,312.5,625	31.3	119.2	2.2	0.7
FB_3_	15.6, 31.25, 312.5, 625	31.3	112.4	0.2	0.1
AFB_1_	6.25, 12.5, 125, 250	12.5	92.2	0.1	0.04
AFB_2_	6.25, 12.5, 125, 250	12.5	109.3	0.1	0.02
AFG_1_	6.25, 12.5, 125, 250	12.5	132.3	0.2	0.1
AFG_2_	6.25, 12.5, 125, 250	12.5	76.8	0.3	0.1
AME	31.25, 62.5, 625, 1250	62.5	89.6	1.1	0.3
HT-2	31.25, 62.5, 625, 1250	62.5	117.7	0.5	0.2
DON	31.25, 62.5, 625, 1250	62.5	112	8.3	2.5
3-ADON	31.25, 62.5, 625, 1250	62.5	127	0.2	0.1
15-ADON	31.25, 62.5, 625, 1250	62.5	97	5.7	1.7

3-ADON: 3-Acetyldeoxynivalenol, 15-ADON: 15-Acetyldeoxynivalenol, DON: Deoxynivalenol, α-ZEL: α-Zearalenol, ZEN: Zearalenone, β-ZEL: β-Zearalenol, FB_1_: Fumonisin B_1_, FB_2_: Fumonisin B_2_, FB_3_: Fumonisin FB_3_, T-2: T-2 toxin, HT-2: HT-2 toxin, AFB_1_: Aflatoxins B_1_, AFB_2_: Aflatoxin B_2_, AFG_1_: Aflatoxin G_1_ and AFG_2_: Aflatoxin G_2_, AME: Alternariol monomethyl ether.

**Table 2 toxins-11-00452-t002:** Overview of mycotoxins levels in poultry feed samples from South Africa.

Analyte ^a^	^b^ % Frequency of Positive Samples	^c^ % Number of Positive Samples above ≥ LOQ	^d^ Mean conc. (µg/kg)	Max. conc. (µg/kg)
ZEN	100	99	71.2	428.9
α-ZEL	100	99	5.4	19.9
β-ZEL	100	99	3.8	22.1
T-2	100	100	3.1	15.3
FB_1_	100	100	1075.6	7125.3
FB_2_	100	100	28.5	125.1
FB_3_	100	100	22.2	115.1
Average FUMs	100	100	375.42	2455.2
AFB_1_	93	98	0.2	0.9
AFB_2_	100	100	0.4	7.1
AFG_1_	97	97	0.7	5.2
AFG_2_	78	82	0.5	1.6
Average AFs	92	97	0.5	3.7
HT-2	100	100	1.9	5.9
AME	100	100	23.1	155.5
DON	99	98	37.8	154.0
3-ADON	98	95	1.6	12.9
15-ADON	59	35	8.9	44.9

3-ADON: 3-Acetyldeoxynivalenol, 15-ADON: 15-Acetyldeoxynivalenol, DON: deoxynivalenol, α-ZEL: α-zearalenol, ZEN: Zearalenone, β-ZEL: β-zearalenol, FB_1_: fumonisin B_1_, FB_2_: fumonisin B_2_, FB_3_: fumonisin FB_3_, T-2: T-2 toxin, HT-2: HT-2 toxin, AFB_1_: aflatoxins B_1_, AFB_2_: aflatoxin B_2_, AFG_1_: aflatoxin G_1_, AFG_2_: aflatoxin G_2_ and AME: alternariol monomethyl ether; ^a^ OTA was analysed but not detected in any of the analysed samples; ^b^ Frequency of positive samples in percentage (in total 105 poultry feed were analysed); ^c^ Number of positive samples above LOQ in percentage; ^d^ Mean concentration levels of mycotoxins in positive samples.

**Table 3 toxins-11-00452-t003:** Mycotoxins contamination level (µg/kg) in poultry feeds per province.

	Gauteng (45)			KwaZulu-Natal (26)			Eastern Cape (5)			Western Cape (17)			North-West (12)		
Analytes	Mean	%F	Max.	Mean	%F	Max	Mean	%F	Max	Mean	%F	Max	Mean	%F	Max
15ACDON	5.9	71	44.9	7.5	69	25.5	3	20	11.8	1.6	35	9.3	4.5	25	30.5
3ACDON	1.4	100	4.7	0.9	96	2.5	2.9	100	7.4	2.1	100	4.5	2.2	92	12.9
DON	43	100	154	23.8	96	81.4	45.9	100	75.8	34.8	100	86.6	46.1	100	137.8
ZEN	102.3	100	429	60.1	100	188.1	93.3	100	347	16.8	100	156.7	46.1	100	187.1
α-ZEL	6.4	100	20	5.3	100	12.2	4.8	100	6.7	4.2	100	13	4.1	100	11.4
β-ZEL	2.1	100	10.9	4	100	18	4.3	100	11	8.3	100	22.1	3.1	100	9.6
FB_1_	1096.8	100	3904.7	959.7	100	3019.6	206.6	100	496.8	3604.4	100	7125.3	1638.2	100	3507.3
FB_2_	19.7	100	72	17.5	100	61.9	4.9	100	13.8	72.8	100	125.1	32.6	100	80.7
FB_3_	15.1	100	69.3	15.2	100	61.4	4.7	100	13.3	55.6	100	115.1	23.9	100	74.4
T-2	3	100	15.3	2.8	100	10.4	4.3	100	11.1	3.8	100	134	2.6	100	5.5
HT-2	1.8	100	5.9	1.7	100	4.2	1.8	100	3.1	2.8	100	5.8	2	100	4.2
AFB_1_	0.2	100	0.5	0.3	100	0.9	0.1	80	0.2	0.1	82	0.3	0.1	75	0.3
AFB_2_	0.6	100	7.1	0.3	100	3.4	0.1	100	0.3	0.3	100	1.2	0.2	92	0.4
AFG_1_	0.8	93	3	0.7	100	1.8	1.4	100	5.2	0.4	82	3.1	0.5	100	2.5
AFG_2_	0.5	100	1.3	0.6	89	1.6	0.3	100	0.7	0.2	10	1	0.2	58	1.6
AME	27.7	100	93.6	32	100	155.5	167	100	70.1	6.1	100	52.2	13.2	100	54.7

(Total number of analysed samples); 3-ADON: 3-Acetyldeoxynivalenol, 15-ADON: 15-Acetyldeoxynivalenol, DON: deoxynivalenol, α-ZEL: α-zearalenol, ZEN: Zearalenone, β-ZEL: β-zearalenol, FB_1_: fumonisin B_1_, FB_2_: fumonisin B_2_, FB_3_: fumonisin FB_3_, T-2: T-2 toxin, HT-2: HT-2 toxin, AFB_1_: aflatoxins B_1_, AFB_2_: aflatoxin B_2_, AFG_1_: aflatoxin G_1_ and AFG_2_: aflatoxin G_2_, AME: alternariol monomethyl ether; %F: Percentage frequency of contaminated samples.

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
