# Peer review of "Analysis of Mycotoxins Contamination in Poultry Feeds Manufactured in Selected Provinces of South Africa Using UHPLC-MS/MS"

_toxins, 2019, doi:10.3390/toxins11080452_

Round 1

Reviewer 1 Report

The authors have made all the suggested changes. For this reason, the manuscript is acceptable in the present form.

Author Response

Thank you very much.

Reviewer 2 Report

No comments

Author Response

Thank you very much.

Reviewer 3 Report

Line 10: The term ppb should not be used, change to µg/kg throughout

Line 44: Aspergillus is repeated twice in the sentence; do you mean Penicillium and Alternaria?

Line 100: The recovery should not be expressed in ppb, but in percentage.

Table 1. The column on R2 is useless as the regression coefficients of calibration curves made with pure standards in solvents are always very high. In turn, the limits of detection (LOD) or quantification (LOQ) would be fine for the table. The calibration levels for mycotoxins are dispensable in the Table and they are mostly unrelated to the LOD and LOQ values in Table 2. For instance, if the lowest calibration level for aflatoxins was 6.25 µg/kg (as reported in Table 1), how could you achieve a LOD of 0.04 µg/kg  (as reported in Table 2)? The apparent recovery should be expressed in percentage (not as ppb). What is instrument accuracy (not defined anywhere)? For OTA, both the Apparent recovery (ppb) and Instrument accuracy (%) are ND (not detected), what do you mean by this? Even if they were no positive samples for OTA the percent recovery and other method performance values should be known.

In my view, the mistakes in calibration levels, in the expression units for recovery and in the terminology generates many doubts about the solvency and quality of analytical results.

Line 113: The findings ‘show’, they do not ‘demonstrate’

Lines 119-120: What do you mean by the sentence ‘The incidence of the different poultry feed groups were in correspondence to the levels of protein and fibre input as presented in Appendix Table 1’

Table 2: Column names should be clarified. What do you mean by %N and %N LOQ?

Table 3: Column name %F, has the same meaning as column name %N used in Table 2?

Lines 181-182: EU has guidance values (not legislation limits) for products intended for animal feed, including feed materials and compound feed for poultry

The first conclusion is that the levels of proteins in poultry feeds along with fibre content are most likely to be the cause of these high values of contamination in poultry feed. However, this is not discussed in the manuscript; the reader is only referred several times to Appendix Table 1. If this is important, it should be analyzed and discussed in the manuscript. However, the levels of fibre and protein may be just indicators of certain ingredients, which would be the reason behind the high mycotoxin levels.

Line 309: Figure 1 is repeated, one Figure 1 is on mycotoxin co-occurrence and the other is on sampling location

Line 345: The dilute and shoot method is fine as long as the matrix effect is measured since it may dramatically affect the method performance (sensibility, recovery, precision). There is no mention of matrix effect on the manuscript and doubts are raised on the validity of analytical results.

Heading 5.5. The mycotoxin levels in calibration curves are not coherent with the LODs and LOQs, especially for the aflatoxins

Line 382: In the recovery experiments, indicate the final levels for each mycotoxin in the spiked samples.

Round 2

Reviewer 3 Report

The explanation on how such very LOD and LOQ values were achieved is not convincing: the calibration levels and spiking levels used do not guarantee the correct calculation of LOD/LOQ even if using the signal-to-noise ratio. Values for precision (RSD% for repeatability and reproducibility) are missing.

The explanation that matrix-matched calibration was used is not convincing: essential details of the procedure are missing (it is not mentioned in the manuscript, anyway) as well as the influence of matrix effects on recovery, sensibility and precision (quantitative and qualitative data are missing)

Again, there is no discussion on the (possible) relationship between protein & fibre content and mycotoxin levels: individual ingredients were not analyzed for. Just a general sentence is not enough.

I reaffirm that the incoherency in calibration and spiking levels, the missing RSD% values, the undefined methodology for matrix-matched calibration and the lack of essential quantitative and qualitative data generates doubts about the solvency and quality of analytical results.

Author Response

Please see the attached. Thank you. 

This manuscript is a resubmission of an earlier submission. The following is a list of the peer review reports and author responses from that submission.

Round 1

Reviewer 1 Report

The manuscript reports on cooccurrence of mycotoxins in poultry feed from SA. The data are of general interest for the readers, organised in a clear way and obtained under sound analytical methodologies. Therefore, I consider this manuscript as worth of publication.

However, my concerns are mainly about the way data are presented and commented. Once reported in tables (which are clear and useful), data should be commented in a more harmonised and structured discussion, considering possible additive/synergistic effects (only shortly touched) under a risk assessment perspective.

Sentences are short and very simple, which is good for English clarity from non-native speakers, but a little too oversimplified. Some more elaborated wording would be an improvement in terms of impact of the work.

I would thus suggest to revise accordingly the discussion (and some parts of the introduction), to change a simple data collection into a more elaborate data interpretation.

Reviewer 2 Report

Quantitative Analysis of Mycotoxins Contamination in Poultry Feeds

The manuscript is related to the natural occurrence of mycotoxins commonly detected in feedstuffs and others less reported in scientific literature. For this purpose, a ultra-high-performance liquid chromatography coupled to a triple quadrupole mass spectrometer has been applied and validated.

The manuscript with the current content is well written and can be published in Toxins, although in my opinion, it is necessary to clarify some essential content of relevance for the reader.

I would like to summarize the main drawbacks of the present work to be considered for publication:

Abstract and introduction

Mycotoxin contamination in feedstuffs commonly reflects the mycotoxin occurrence in raw materials and ingredients employed in feed manufacture, as mentioned in introduction section. However, does the authors now the raw materials employed in feed formulation? It would be useful to compare the contents between feedstuffs with different composition. In the discussion section it is said that maize-based products are commonly contaminated by Fusarium mycotoxins, mainly FUM. As FUM have been detected in 100% of samples, it would be important to relate this fact to the composition of analysed samples.

Line 18. Why is this surprisingly?

Line 43. Health effects on animals, especially in poultry can be detailed.

Line 50. References 8 and 9 can be described better. The mycotoxin occurrence in these studies must be described.

More detailed information on the carry-over must be provided. A reference on mycotoxin carry-over into edible parts and by-products must be added.

Introduction is scarce, more information, especially on those less reported mycotoxins, such as Alternaria toxins and ZEN derivatives must be providen.

Results

Line 61-64. This information is not related to mycotoxin identification, in my opinion it should be included in sampling section.

Table 1. Please abbreviate retention time as RT.

Table 1 to 5. In footnotes FB1 is Fumonisin B1 in singular, not in plural.

Line 80. What does “ready-made” means in this context?

The mycotoxin occurrence is different depending on the group is intended for (breeders, growers and finishers). Could be this fact related to feedstuff ingredients?

Table 4. The format in some columns is not alligned.

Discussion

Line 128. Please, describe the specific composition of analysed samples.

Lines 141-151. The difference in mycotoxin contents between colder and warmer climates is not well defined. Information is confusing for the reader.

Line 147. ...analysed previously...” It is not clear the samples the authors refer to.

Lines 155-157. What is the difference between the first DON frequency and the second one? The second one is detected in South African chicken feed, and the first one?

Line 159. Please, indicate the levels detected in references 3 and 18, thus is easier for the reader to compare the results.

Line 168. In my opinion the term “scarce” is more suitable than “inadequate”.

Line 177: Aspergillus parasiticus instead of paracitucus.

Line 181: …to monogastrics of pig and poultry. Change into: …to monogastrics such as pig and poultry.

Line 194: Please, give more detail on mycotoxin co-occurence.

Conclusions

Some paragraph related to Alternaria mycotoxins should be included, as they have been detected in analysed samples.

Material and methods

Sample section must be improved. Information on sample composition is scarce.

Line 241. Reference number 41? There are 36 references.

Lines 262 and 268. quantitation or quantification?

Which type of samples have been used for calibration purposes?

One of the main objectives of the study is to develop a new analytical method, but there are scarce data about validation study, for instance values of precision and repeatability for each mycotoxins are not indicated.

Reviewer 3 Report

The title is quite large and vague. Please state on the place (or country) and the method used should  be already specify in the title. There are many quantitative analytical methods. Place and

The analytical method was not checked against any reference material ?

Are the results corrected for recovery data provided in the manuscript ?

Throughout all the manuscript, please check carefully all the numbers and percents and round them with one decimal place (ex. 38.7% instead of 38.74%),  

No correlation analysis has been tested between the levels of co-occurred mycotoxins. Please perform this statistical analysis in order to check the consistency of the relationships between the levels of co-contamination of different mycotoxins. Table of correlation coefficients between different mycotoxins should be derived and presented.

Please present the co-contamination results.

Table 1 should be presented as supplementary data.

For official control, the non-compliant samples or batches should be declared to the health authorities.  Please state on this requirement and possible actions taken onwards.

Taking into account the maximum level of each mycotoxin, please check and discuss the levels against the legislation (local or international maximum accepted limits) or the consumption of these feeds with the LD50 doses found in the literature. The consequences for poultry involved as well as for consumers of the meat or offal derived from these feedings ?

Reviewer 4 Report

General Comments

This manuscript describes analysis of poultry feeds sampled in South Africa. A number of problems are apparent.

The authors should NOT be using ppb as concentration units, but should know that scientific results are always reported in SI units.

As chemical analysts the authors seem unaware of the meaninglessness of their use of so many significant figures for their results. Their results should have only 3 or 4 significant figures. They need to consider that they state the sample used was 10g, presumably between 9.9 and 10.1, i.e. 3 significant figures.

The results seem to indicate that all the poultry feeds meet SA standards for the mycotoxins that are regulated. This is surely good news, but doesn’t feature in their abstract or conclusion. As an aside, SA feed manufacturers generally analyse all feed components as feeds that fail can result in large economic losses. For this reason, the second sentence of their key contribution is really not valid as manufacturers are only too aware of mycotoxin problems.

Specific Comments

Line 14: The words Alternaria toxin are left hanging.

Line 18: Why surprising?

Line 21: The birds are exposed to toxin, not the toxin to the birds as written here.

Line 33: Avoid twice using the word significant.

Line 48: Correct the English in These study was .

Line 65: It is not clear what the authors mean by dilute method. Usually this type of method is referred to as dilute-and-shoot, but their experimental doesn’t mention dilution of the extract.

Table 2: This table is not mentioned until the end of the manuscript so appears out of place here.

Table 2: The column calibration level contains 4 figures. It is not clear how they relate to the 5 concentrations mentioned in their section 5.5 on validation.

Table 2: What is the last column, Instrument accuracy?

Table 2: Most surprisingly, for OTA, FB1 and DON, the apparent recovery is given as ND, meaning not detected. How is this possible? To me it implies the results for these 3 toxins are invalid and the manuscript flawed.

Section 2.2 (Tables 4 and 5): Are results mentioned here and across the rest of the manuscript corrected for recovery?

Table 4: I failed to see superscript d in the body of the table.

Table 4: There is a disconnect between %N and N, which I assume is due to trace levels above LOD and below LOQ, but this is not stated. As it stands, it seems the authors can’t calculate percentages.

Table 4: The main problem with table 4 lies in the FB results. At face value, their total FUM (FB1+FB2+FB3) is markedly less than the FB1 level! The FBs in feeds are naturally occurring and hence should always be in a range of about 70% FB1 of the total. This may vary up and down, but these results show it to be of the order 95%, which means the analysis is wrong. Compounding this problem is that the LOD and LOQ quoted in the last columns show good values for FB2 and FB3, but those for FB1 are orders of magnitude higher and again point to analytical problems.

Table 5: Is %F the same as %N in Table 4?

Line 130: ZEN has long been known as a contaminant of cereal crops!

Line 143: There is no such thing as “acrylated derivatives”, the authors mean acyl derivatives or more correctly acetylated.

Lines 168-169: The setting of regulations results from toxicological studies, not the presence in human tissue.

Section 5.1: There seem to be 5 sampling sites, so where multiple samples of a particular feed were taken, were they of different batches?

Line 245: Give g-value, not rpm.

Line 246: Dilution is mentioned in section 2.1, but not here?

Line 262: Electro-spray, not electron spray.